# Insulin and Oral Hypoglycemic Drug Overdose in Post-Mortem Investigations: A Literature Review

**DOI:** 10.3390/biomedicines10112823

**Published:** 2022-11-05

**Authors:** Alice Chiara Manetti, Giacomo Visi, Federica Spina, Alessandra De Matteis, Fabio Del Duca, Emanuela Turillazzi, Aniello Maiese

**Affiliations:** 1Department of Surgical, Medical, and Molecular Pathology and Critical Care Medicine, Institute of Legal Medicine, University of Pisa, via Roma 55, 56126 Pisa, Italy; 2Department of Anatomical, Histological, Forensic and Orthopedical Sciences, Sapienza University of Rome, Viale Regina Elena 336, 00161 Rome, Italy

**Keywords:** insulin, oral hypoglycemic drug, overdose

## Abstract

*Background and Objectives:* Insulin and oral hypoglycemic agents are drugs widely used in the world population due to their therapeutic effects on diabetes mellitus. Despite these benefits, they can also cause accidental or voluntary drug overdose. This review aims to evaluate post-mortem investigations in cases of suspected hypoglycemic drug overdose. *Materials and Methods:* We performed a comprehensive search using the Preferred Reporting Items for Systematic Review (PRISMA) standards; we systematically searched the PubMed, Science Direct Scopus, Google Scholar, and Excerpta Medica Database (EM-BASE) databases from the point of database inception until August 2022. The following inclusion criteria were used: (1) original research articles, (2) reviews and mini-reviews, (3) case reports/series, (4) and only papers written in English. *Results:* Thirty-three scientific papers, including original research articles, case reports, and case series, fulfilled the inclusion criteria. A total of 109 cases of insulin or hypoglycemic drug overdose were found. There were 71 cases of suicide (65%), 25 cases of accidental poisoning (23%), and 13 cases of homicide (12%). The most commonly used drug was insulin (95.4%). Autopsy and post-mortem examinations were performed in 84 cases, while toxicological investigations were performed in 79 cases. The most common gross findings in the autopsy were pulmonary edema (55.7%) and congestion (41.8%), while the most common histological finding was neuronal depletion or necrosis (29.1%). *Conclusions:* In the suspicion of death from insulin or overdose from oral hypoglycemic agents, autopsy findings may be nonspecific, and the search for injection marks can be positive at the external examination. Significant post-mortal alterations can interest biological samples and an early autopsy investigation is recommended.

## 1. Introduction

Insulin and oral hypoglycemic agents (OHA) are widely used due to the high incidence of diabetic disease in the general population. However, despite their widespread use, they rarely are involved in overdose episodes. According to the *2015 Annual Report of the American Association of Poison Control Centers*, out of a total of 2,168,371 people exposed to drug poisoning, only 6880 took insulin, and only 14,534 took OHA [1]. Jefferys and Volans show that in a total of 386 poisoned diabetic patients, suicide rates are around 4.7% for insulin and 12% for oral antidiabetics [2]. On the other hand, self-intentional overdose is less frequent in the case of non-diabetic patients, in which around 1% use insulin [3]. Nevertheless, it is well known that hypoglycemic drugs are also used by non-diabetic patients for suicidal purposes. In particular, healthcare personnel and diabetic patients’ relatives, thanks to their position, can easily access these drugs and use them to attempt suicide or commit homicide [4]. As far as the forensic investigation is concerned, the post-mortem diagnosis by antidiabetic overdose is particularly difficult. This is due to the absence of pathognomonic findings at the autopsy. Thus, in many cases, the diagnostic gold standard remains the biochemical detection of insulin in post-mortem samples, even if it is not easy to achieve due to the very fast metabolism of insulin [5].

Herein, a review of the literature on hypoglycemic deaths due to insulin and OHA overdose is presented. The included papers regard cases from various countries around the world, although most are from Western countries. The aim of this work is to highlight the autopsy findings and the fundamental toxicological examinations in such cases. This review can help forensic pathologists to consider hypoglycemic drug overdose when the cause of death is not clear and to perform all the useful investigations.

## 2. Materials and Methods

The present systematic review was carried out according to the Preferred Reporting Items for Systematic Review (PRISMA) standards [6]. A systematic literature search and critical review of the collected studies were conducted. An electronic search of PubMed, Science Direct Scopus, Google Scholar, and Excerpta Medica Database (EMBASE) from database inception until August 2022 was performed. The search terms were “insulin”, “oral hypoglycemic agents”, “hypoglycemia”, “autopsy”, “death”, “overdose”, “suicide”, “homicide”, and “accidental”, and these terms were searched in the title, abstract, and keywords. The bibliographies of all located papers were examined and cross-referenced to further identify relevant literature. A methodological appraisal of each study was conducted according to the PRISMA standards, including an evaluation of bias. The data collection process included study selection and data extraction. The following inclusion criteria were used: (1) original research articles, (2) reviews and mini-reviews, (3) case reports/series, (4) and only papers written in English. Non-English papers, papers regarding cases of survival after drug-induced hypoglycemia, and papers about deaths due to biguanide (an OHA) overdose were excluded. Concerning the papers involving biguanide overdose, they were excluded because the lethal mechanism is not related to hypoglycemia.

Three researchers (G.V., F.S., and F.D.D.) independently examined the papers with titles or abstracts that appeared to be relevant and selected those that analyzed deaths due to overdose of insulin and/or OHA. Disagreements concerning eligibility among the researchers were resolved by consensus. Preprint articles were included. Data extraction was performed by two investigators (A.C.M. and A.M.) and verified by other investigators (E.T. and A.D.M.). This study was exempt from institutional review board approval, as it did not involve human subjects.

## 3. Results

The search identified 203 articles, which were screened to exclude duplicates. The resulting 198 references were then screened based on their title and abstract, which left 93 articles for further consideration. These publications were carefully evaluated considering the main aims of the review. This evaluation left 33 scientific papers comprising original research articles, case reports, and case series. Figure 1 illustrates our search strategy. The included papers involve cases from various countries around the world, although most were from Western countries. No paper from Africa was found.

In Table 1, a brief description of the studies included in this review is reported.

We found a total of 109 cases of insulin or hypoglycemic drug overdose. As shown in Table 2, which summarizes the main characteristics of the articles included in this review, 63 (57.8%) were male while 46 (42.2%) were female. The age of the subjects was specified in only 56 cases. Among these cases, the most impacted was the >50-year-old population (48.2%). There were 71 cases of suicide (65%), 25 cases of accidental poisoning (23%), and 13 cases of homicide (12%), as shown in Figure 2. The most commonly used drug was insulin (95.4%). At least 58 cases regarded people who suffered from diabetes mellitus (53.2%). Among suicide cases, 14 subjects were health care workers (19.7%), while among homicide cases, the killer was a health care worker in 5 cases (38.5%). In most cases, the drug was administered subcutaneously (77.1%) and the preferred site of injection was the abdomen (24/84 cases). This is highlighted in Table 3.

Toxicological investigations were performed in 79 cases (76 insulin overdoses, 1 OHA overdose, and 2 insulin plus OHA overdose cases). Table 4 reports the different types of analyses that were performed. As shown, the most commonly performed analysis when insulin overdose was suspected was the quantification of insulin in blood (70.5%), while the next most common analysis was C-peptide blood quantification (39.7%). Vitreous humor glucose was quantified in 20 cases (25%), while the insulin vitreous humor levels were quantified in only 5 cases (6.4%).

Autopsy and post-mortem investigations were performed in 84 cases (79 insulin overdoses, 3 OHA overdoses, and 2 insulin plus OHA overdose cases). In Table 5, a summary of the main post-mortem findings is shown. The external examination didn’t show any peculiar finding, except for the injection mark (67.1%), when the administration is through a needle. Regarding the autopsy, it showed unremarkable features; the most common gross findings were pulmonary edema (55.7%) and congestion (41.8%), while the most common histological finding was neuronal depletion or necrosis (29.1%).

## 4. Discussion

Diabetes represents a widespread pathology in the modern world. The *Global Report on Diabetes* (WHO, 2016) shows that in 2014, approximately 422 million adults suffered from diabetes. As a result, insulin and other OHA are widely used by the population. In particular, insulin is prescribed with greater frequency in type 1 diabetic patients, although it is also used in some cases of type 2 diabetes, namely those inefficiently controlled by other drugs.

Hypoglycemia is a possible complication of hypoglycemic drugs (especially insulin), and if not correctly diagnosed and treated, it can lead to death. Hypoglycemia manifests as a procession of symptoms grouped in the so-called Whipple’s triad, which is characterized by symptoms likely to be caused by hypoglycemia, low blood glucose concentration, and relief of symptoms if blood glucose levels increase [40]. Symptoms and signs of hypoglycemia may be divided into neurogenic and neuroglycopenic. Among the former, there are warning symptoms caused by autonomic stimulation (e.g., tremors, sweating, hypersalivation, tachycardia, etc.) that can be solved by swallowing simple sugars to increase blood glucose. Neuroglycopenic symptoms are characterized by a reduction in cognitive function and are less responsive to corrective measures. If untreated, hypoglycemia may induce a state of restlessness, violent behavior, unconsciousness, cerebral seizures, or even coma. Normally patients do not die during hypoglycemic episodes, even if the episodes are severe [41]. In the rare cases in which death occurs, it can be sudden or the comatose state can lengthen the timing of its realization. The exact mechanisms by which hypoglycemia causes sudden death is not yet well understood, but arrhythmias and cerebral seizures seem to play a major role [42]. Furthermore, a clear diagnosis is particularly difficult even when autopsy and toxicological examinations are performed. This work shows that among these kinds of deaths, the most common are suicide (65%), while the remainder are events that occurred accidentally (23%) or intentionally as acts of homicide (12%).

Compared to the wide use of insulin and oral hypoglycemic agents, fatal overdoses are relatively limited in the literature. In the home setting, dosing errors in self-administration of insulin and OHA can lead to hypoglycemic events. Such errors occur much more frequently in insulin therapies. Sometimes high self-administered dosages lead to very severe hypoglycemia with fatal consequences. Intentional overdoses with these drugs for suicidal purposes are more frequent among diabetic patients, as our review showed. Indeed, among the cases reported in our work, out of a total of 71 suicides, 37 subject were diagnosed with diabetes mellitus.

It is important to remember that diabetes is a disease that involves different systems, including the nervous system, such that patients affected by this pathology have double the risk of developing depressive disorders as compared with the general population [43]. However, in some cases of suicidal overdose in diabetic patients, simultaneous depression has not been recognized. Therefore, such cases may be classified as accidental [32]. This can lead to inaccuracy in the results found in the literature.

Antidiabetics are considered “silent weapons” as they produce non-specific post-mortal alterations that sometimes mimic natural deaths. As previously stated, a clear diagnosis of a fatal overdose is particularly difficult if circumstantial data, autopsy examination, or the toxicological data are lacking.

Another peculiar aspect highlighted in this work concerns the high incidence of the use of these drugs for suicidal or homicidal intent among health care personnel. Concerning suicides, this work shows that out of 71 cases, 14 were performed by people who worked in health care. Such people have easy access to these drugs and knowledge of their mode of action. The observations reported by Bugelli et al. and Behera et al. are very peculiar [31,35]. In the first report, the suicide victim was a former paramedic who chose the sublingual route of administration, while in the second report, the suicide victim, a physician, used the intravenous route. As Behera et al. specify, the subject knew that intravenously, insulin bioavailability would have been 100% and that a bolus dose of the drug would have killed him in a very short time [31]. In both cases, the two subjects, due to their medical knowledge, chose two ways of administering insulin with a faster rate of action. The easy access to these substances by health personnel, as well as knowledge of their fatal potential, also extends to murders. Among the 13 cases of homicides detected in our work, 5 cases contained murderers who worked in the healthcare system (23.1%). In particular, the case presented by Birkinshaw et al. is the first case of insulin murder reported in the literature, and the killer was a nurse [7]. In the case presented by Uezono et al., the perpetrator was a pharmacist who killed his offspring by administering insulin, glibenclamide, and benzodiazepines [25]. The murder presented by Iwase et al. was performed by an assistant nurse, who administered insulin and air intravenously [24]. Finally, two of the four cases presented by Tong et al. were perpetrated by nurses. One injected insulin into the victim while lying that it was a dose of a vaccine; the other injected insulin into her fiancé after she had deceived him into drinking water laced with benzodiazepines [33]. In all these cases, the hypothesis was that the deceased, who were mainly not diabetic, experienced severe hypoglycemia that caused the death.

If there are not many data in the literature on deaths from insulin or OHA overdose, even more limited are the data regarding autopsy investigations in such cases. However, some considerations can be formulated based on the available data. Starting with the external inspection, particular attention must be paid to the search for injection marks. Although difficult, such research can often be positive. As shown by the cases examined in our work, injections can be performed on various locations of the body, especially the abdomen. Regarding the gross examination, we found that the most reported findings are pulmonary edema and congestion along with neuronal depletion or necrosis at the histological analysis. These findings are unspecific, and therefore they could not be considered pathognomonic to diagnose hypoglycemia deaths.

Watkins and Thomas note that deaths due to hypoglycemia show brain injuries. Moreover, these changes primarily concern the temporal lobes and the hippocampal cortex [41]. Additionally, Tong et al. note a conspicuous involvement of the encephalic district [33]. Stephenson et al., in their large series of cases, specify that neuronal depletion, if massive, is generally observed in deaths from hypoglycemia with a prolonged survival period between injection and death. [39] This could be interpreted as a sign of prolonged hypoxia.

Based on our data, we can state that an overdose of insulin or OHA does not produce pathognomonic changes; often the findings are unspecific and could be detectable even in natural deaths. Certainly, it is clear that one of the most affected organs is the brain, which experiences serious neuronal necrosis [44].

To facilitate the post-mortem investigation in cases of suspected death due to insulin or OHA overdose or death from suspected hypoglycemia, we provide a flowchart that could help forensic pathologists (Figure 3). First of all, a thorough crime scene investigation needs to be performed since it may provide important information. The forensic pathologist must carefully examinate the location and the corpse and try to collect all the following circumstantial data:Location: if syringes, vials, injector pens, or tablets are present, he or she should demand appropriate toxicological investigation of their contents;Corpse: the first examination of the body should evaluate the presence of any injection sites, paying attention to areas such as the abdomen, thighs, and buttocks region. This research can be negative as the syringe used for the insulin generally has a high gauge needle;Circumstantial data: information on the victim’s medical history (diabetes, depression, etc.), the presence of diabetics in the family treated with insulin or OHA, previous suicide attempts, drug addictions, and occupation (e.g., health worker) should be collected.

If some of these factors are present, death due to drug-induced hypoglycemia can be reasonably suspected. At the external examination, if not found during the crime scene investigation, the injection sites should be searched again. If found, they must be excised for subsequent histological, immunohistochemical, and toxicological analyses. Unfortunately, the absence of evident pathognomonic data, revealed by this review, limits suggestions for specific research to be conducted at the internal examination. Nevertheless, sampling of the different matrices (blood, liquor, vitreous humor, urine, and other organs) is fundamental when carrying out the toxicological investigations.

For a certain diagnosis, in addition to a thorough gross investigation and microscopic examination of the histological samples, the crime scene investigation is of paramount importance. Numerous valuable pieces of information can be detected at the crime scene, including the presence of drugs, the type and quantity of drugs taken, as well as the methods of administration.

Toxicological investigations are fundamental, even if they can be affected by post-mortem and decomposition phenomena. Generally, when suspecting death from insulin overdose, toxicological investigations, in addition to searching for insulin, can evaluate C-peptide and glucose levels. In cases suspecting death due to oral hypoglycemic agent overdose, however, toxicological investigations can limit themselves to the search for the drug and glucose.

One of the most commonly used matrices for such research is blood, as it is one of the most common matrices to be collected during an autopsy. It is well known that insulin and C-peptide detection and quantification in post-mortem samples are characterized by important analytical limitations. Insulin degradation occurs in the corpse after death, and some authors report significant changes in the concentration happening even in a 24 h post-mortem interval [24]. Insulin is unstable in post-mortem blood samples as well, and this is due to the degradation of insulin by enzymes released during the hemolysis of red blood cells [45,46,47]. However, it has to be considered that insulin concentrations could increase in central blood samples after death, which is hypothesized to result from endogenous insulin diffusion from the pancreas through the portal vein. This is why some authors suggest using femoral blood samples to conduct such investigations [48].

Analytical limitations in the quantification of insulin and C-peptide in blood samples create problems for using the ratio of insulin to C-peptide, an interesting hypothetical value that could be used to distinguish between hypoglycemia due to exogenous or endogenous insulin. Theoretically, an undetectable C-peptide concentration may suggest the absence of endogenous insulin production [32].

Among other biological samples, vitreous humor is the most interesting for this kind of toxicological investigation. It is less vulnerable to diffusion and decomposition effects [49]. However, Coe reported that insulin hardly penetrates the blood-vitreous barrier [50]. Additionally, the glucose concentration in the post-mortal matrices can be altered and therefore may not show the real antemortem value [35,51].

Peripheral blood glucose concentrations drop around 1–2 mmol/L per hour, and in central blood samples, glucose concentration may be increased by enzymatic breakdown of hepatic glycogen, which then gets transferred into large vessels [52]. Generally, a glucose concentration of less than 200 mg/dL in postmortem peripheral blood samples is considered normal [53].

The effects of alterations of oral hypoglycemic agents in post-mortal matrices is less well known.

Some authors suggest paying particular attention to the fact that death from hypoglycemia can occur even after a prolonged comatose state, a period that can allow the elimination of insulin, its metabolites, and other substances. [53].

Traditional methodologies utilizing radioimmunoassays and enzyme-linked immunoassays are still the most commonly employed strategies for the investigation of hypoglycemia. However, these methods show evident limits in the identification, differentiation, and quantification of the different drugs [54]. Immunoassays investigating insulin analogues show cross-reactivity, and this is considered an important limitation in the effective quantification and identification of the substance [55,56,57].

However, mass spectrometry, a more specific method for distinguishing substances, has experienced greater utilization in recent years [58]. This method allows discrimination between different analytes such as human insulin, degradation products, metabolites, and insulin analogues. [57]

Moreover, other promising methods, such as liquid chromatography with tandem mass spectrometry (LC–MS-MS) have recently been demonstrated to perform well in both quantification and differentiation between insulin analogues [29,30,47,50].

Another peculiar observation made in this review pertains to the toxicological or histochemical investigation of injection site tissues, as some authors reported deposits of insulin in these areas [59].

Injection sites serve as a reservoir where traces of unabsorbed insulin may be found even days after the injection. This persistence obviously depends on the dose, type of insulin, and other factors [33,52]. Attention must be paid to diabetic patients, with whom a positive result of this kind of investigation can only determine if the type of insulin is the one prescribed to the subject or not [57].

## 5. Conclusions

This work reviewed 109 cases of death due to insulin and oral hypoglycemic agent overdose. It showed that when a forensic pathologist is asked to evaluate a suspected death due to insulin or OHA overdose, a certain diagnosis may be difficult to be carried out. The investigation must include a thorough crime scene evaluation, and the autopsy has to be followed by specific toxicological analyses. During a crime scene investigation, the pathologist must pay attention to all the elements that can support the hypothesis of death due to insulin or OHA overdose. These include the presence of syringes, vials, injector pens, or tablets as well as the presence of injection marks on the corpse. Circumstantial data may provide essential information too. During the autopsy investigation, the macroscopic and microscopic findings can be unspecific.

On the other hand, toxicological investigations [60,61] may be remarkably important, even if they can be affected by post-mortem and decomposition phenomena. It is well known that insulin, C-peptide, and glucose detection and quantification in postmortem samples are characterized by important analytical limitations. This kind of investigation can be conducted on many different matrices utilizing different analytical methods. The finding of injection sites can help the pathologist, especially in the case of a non-diabetic patient. Indeed, toxicological and histochemical investigations can be performed on these tissues, as they function as a reservoir, where traces of unabsorbed insulin may be found. Furthermore, this work underlines the importance of early autopsy investigation to reduce significant post-mortal alteration in the samples to be analyzed.

In conclusion, this review shows that, in the suspicion of death from insulin or oral hypoglycemic agent overdose, autopsy findings may be unspecific and that a certain diagnosis can be carried out on the basis of toxicological investigation integrated with circumstantial data. For this reason, the creation of protocols and guidelines showing the fundamental steps for the certain diagnosis of death due to insulin or OHA overdose should be considered.

## Figures and Tables

**Figure 1 biomedicines-10-02823-f001:**
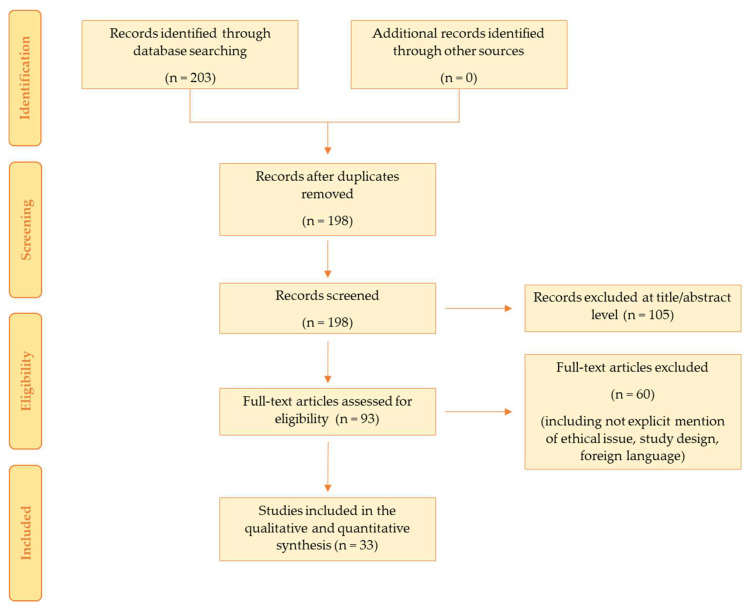
This figure shows our search strategy following PRISMA standards.

**Figure 2 biomedicines-10-02823-f002:**
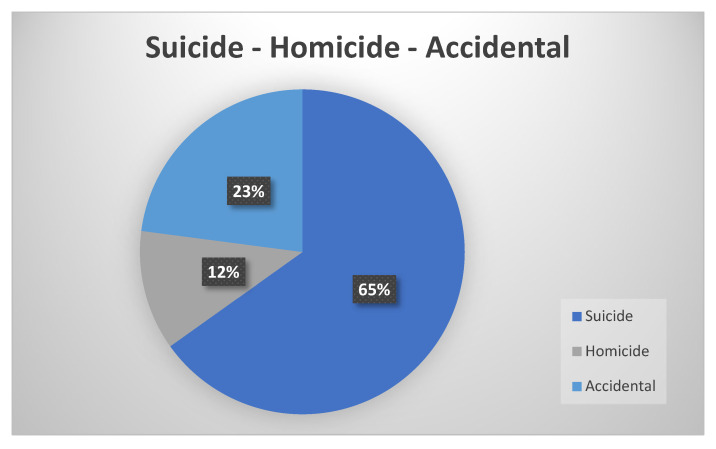
The graphic shows the percentages of the different modes of death in the 109 analyzed cases.

**Figure 3 biomedicines-10-02823-f003:**
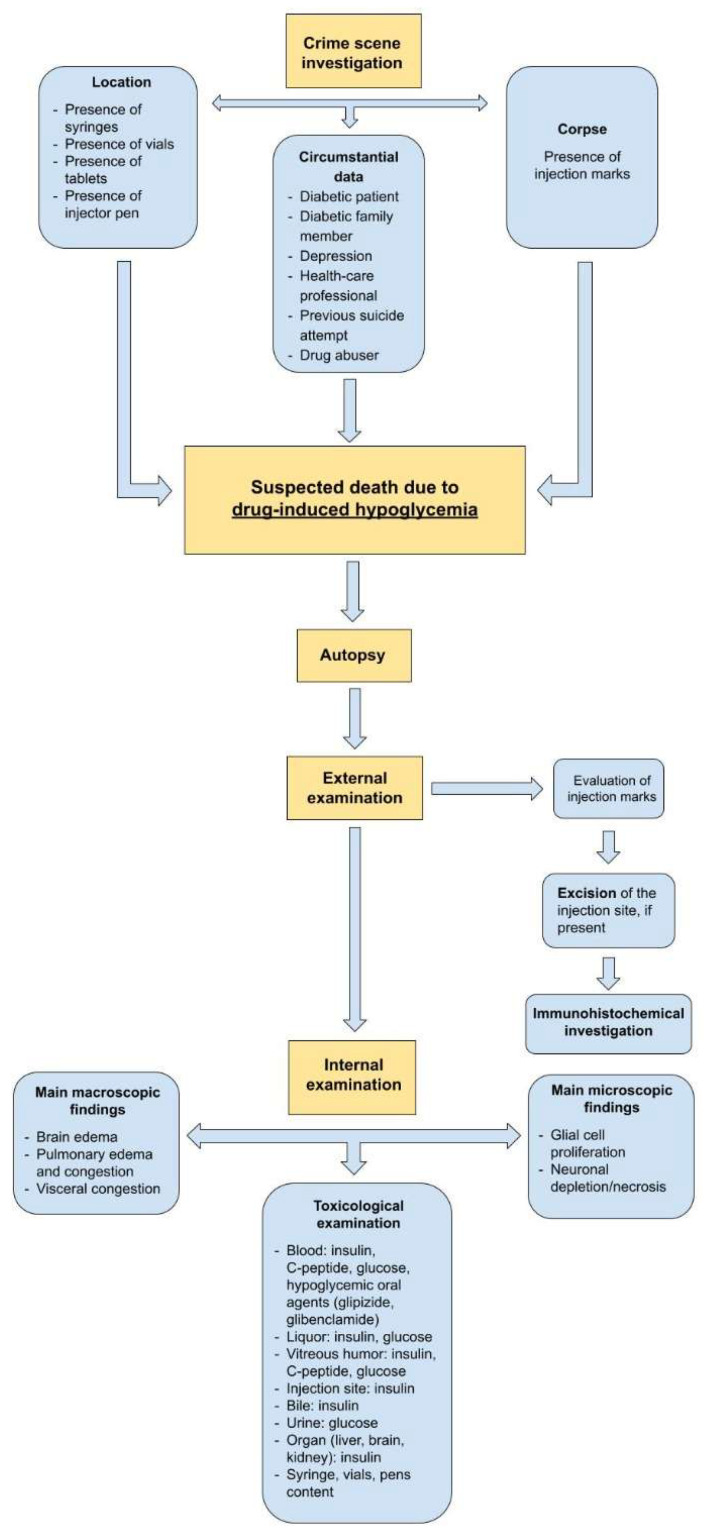
The flowchart shows the major post-mortem investigations we suggest for cases of suspected death due to drug-induced hypoglycemia.

**Table 1 biomedicines-10-02823-t001:** Review of the literature on deaths caused by insulin and OHA overdose.

Reference	n° Cases	Age and Sex	Diagnosis of Diabetes	Drug Used	Mode of Administration	Brief Case Description and Previous Medical Conditions	In Life Laboratory Analysis	Mode of Death	External Examination and Autopsy Data	Post-Mortem Toxicological Analysis	Cause of Death
Birkinshaw et al. 1958 [7]	1	30 F	No	Insulin	SC	Found death in the bath full of water. The husband, which was the killer, worked as nurse.	NA	Homicide	Four IMs on the buttocks; signs of drowning.	↑ insulin levels in the soft tissues under the IMs (84 U in 170 g. of tissue)	Drowning after loss of consciousness due to insulin overdose.
Price 1965 [8]	1	21 M	No	Insulin	SC	Psychopathy traits.	NA	Suicide	IMs at the thigh and chest. Microscopic examination showed perivascular hemorrhages.	↓ blood glucose levels; ↓ liquor glucose levels; ↑ urine glucose levels.	Insulin overdose.
Dickson et Cairns 1977 [9]	1	37 M	Yes	Insulin and nitrazepam	SC	Suffered from depression.	NA	Suicide	IMs at the legs and arm.	↑ blood insulin levels; insulin traces in injection site samples	Insulin overdose.
Hansch et De Roy 1977 [10]	1	27 M	Yes	Insulin	SC	Suffered from depression.	NA	Suicide	IMs at the hand and buttock; bronchopneumonia.	Blood negative for insulin traces; blood glucose levels unreliable.	Bronchopneumonia due to hypoglycemic coma.
Arem et Zoghbi 1985 [11]	2 *	31 F	Yes DM1	Insulin NPH + regular	SC	Died on day 12 of hospitalization.	At ED admission, blood glucose levels were 1.56 mmol/L.	Suicide	NA	NA	Insulin overdose.
		27 M	Yes DM1	Insulin	SC	Died on day 10 of hospitalization.	At ED admission, blood glucose levels were 1.17 mmol/L.	Suicide	NA	NA	Insulin overdose.
Levy et al. 1985 [12]	1	56 M	No	Insulin	IV	Recovered for surgery. Found severely hypoglycemic on the ward bed. Poisoned by wife during a visit. Died on day 3 after poisoning.	Blood glucose levels were 6.9 mmol/L.	Homicide	No remarkable findings.	NA	Insulin overdose.
Grunberger et al. 1988 [13]	1 *	60 M	NA	Insulin	SC	Worked as hospital administrator; suffered from depression; admitted to the hospital for insulin overdose a few months before death.	NA	Suicide	NA	↓ blood glucose levels (0.4 mmol/L); blood amitriptyline levels were 2870 mmol/L.	Insulin overdose.
Patrick et Campbell 1990 [14]	4	35 M	Yes DM1	Insulin	SC	Suffered from alcohol abuse; died on day 17 of hospitalization for bronchopneumonia.	At ED admission, blood glucose levels were 0.6 mmol/L; blood insulin levels were 9940 mU/L.	Accidental	Signs of bronchopneumonia; brain edema with moderate tentorial herniation and necrotic areas; 2-week-old cerebral infarction.	NA	Bronchopneumonia and insulin overdose.
		31 M	Yes DM1	Insulin	SC	Died on day 4 of hospitalization.	At ED admission, blood glucose levels were 12.1 mmol/L.	Accidental	Signs of brain hypoxia and necrotic areas.	NA	Insulin overdose.
		71 M	Yes	Insulin NPH	SC	Died on day 17 of hospitalization; a chest X-ray showed aspiration pneumonia.	At ED admission, blood glucose levels were 1.1 mmol/L.	Accidental	Signs of bronchopneumonia; pancreas adenocarcinoma; bilateral carotid and vertebral arteries atheroma; ACA aneurism; brain edema, neuronal depletion, and 2–3-week-old infarction areas.	NA	Aspiration pneumonia and insulin overdose.
		31 M	Yes DM1	Insulin	SC	Died on day 13 of hospitalization; he also had bronchopneumonia.	At ED admission, blood glucose levels were 1.1 mmol/L.	Accidental	Brain edema; bilateral uncal herniation; hemorrhagic infarctions; hypoxic changes 2 week-old lesions.	NA	Bronchopneumonia and insulin overdose.
Patel et al. 1992 [15]	1	NA, F	No	Insulin	Unspecified	Nurse; suffered from depression (4 previous suicide attempts with insulin).	NA	Suicide	Vomit stains; fingernail cyanosis; intrapulmonary gastric content aspiration with acute inflammation; lung congestion; pleural and epicardial petechiae; brain edema.	↑ blood insulin levels (free insulin 1200 μU/mL; total insulin 1257 μU/mL).	Insulin overdose.
Beastall et al. 1995 [16]	1	33 F	No	Insulin + paracetamol (tablets)	SC	Suffered from alcohol abuse; previous suicide attempt; her insulin-dependent flat challenged her to self-administer insulin.	NA	Induced suicide	IM in the antecubital region of the upper limb.	↓ glucose levels in heart blood (0.3 mmol/L) and VH (about 0 mmol/L); ↑ insulin levels in blood (120 mU/L) and in the soft tissues under the IMs (13.1 mU/g); ↓ C peptide blood levels (0.06 nmol/L)	Insulin overdose.
Peschel et al. 1995 [17]	6 *	28 F	NA	Insulin	SC	Nurse.	NA	Suicide	IM at thigh.	NA	Insulin overdose.
		42 F	NA	Insulin	SC	Nurse.	NA	Suicide	IM at thigh.	NA	Insulin overdose.
		40 M	NA	Insulin + Methohexital	IV	Nurse.	NA	Suicide	IM at the forearm.	NA	Insulin overdose.
		31 M	NA	Insulin	SC	-	NA	Suicide	IM at the forearm.	NA	Insulin overdose.
		55 M	NA	Insulin	IV	Physician.	NA	Suicide	IM at the antecubital region.	NA	Insulin overdose.
		43 F	NA	Insulin	SC	Nurse.	NA	Homicide	IMs at upper arm and buttock.	NA	Insulin overdose.
Lutz et al. 1997 [18]	2	51 M	Yes	Insulin (Actrapid + Protaphan)	SC	Found dead at home.	NA	Suicide	Advanced putrefaction.	NA	Insulin overdose.
		57 F	No	Insulin (Actrapid + Protaphan)	SC	Found dead at home.	NA	Suicide	Putrefaction and beginning mummification; IM at the tight.	NA	Insulin overdose.
Cooper 1998 [19]	1	68 F	Yes DM1	Insulin NPH	SC	She was on insulin treatment; hypoglycemia following a reduced caloric intake period.	Fingerstick blood glucose levels < 1.4 mmol/L.	Accidental	NA	NA	Insulin overdose.
Kernbach et Puschel 1998 [20]	12	42(range 21–61, 4 F and 8 M,	9 yes; 3 no.	Insulin	9 SC; 2 SC + IV; 1 unspecified	Four medical personnel; two relatives of diabetics; eight suffered from depression (at least one previous suicide attempt).	NA	Suicide	IMs in 11 cases; in 10 cases cerebral edema, pulmonary edema, and visceral congestion; in 2 cases no remarkable findings.	For all the cases: glucose and insulin levels in liquor; glucose levels in VH; HbA1c and insulin levels in blood.	Insulin overdose.
Fernando 1999 [21]	2	30 M	No	Glibenclamide	Oral	Several episodes of hypoglycemia.	At ED admission, blood glucose levels were 2.39 mmol/L.	Homicide	No remarkable findings.	NA	Glibenclamide overdose.
		59 F	No	Glibenclamide	Oral	Several episodes of hypoglycemia.	NA	Homicide	No remarkable findings.	NA	Glibenclamide overdose.
Koskinen et al. 1999 [22]	1	48 M	No	Insulin	Unspecified	Vegetative state for two months before the death.	At ED admission, blood glucose levels were 0.3 mmol/L.	Homicide	CAD; laminar necrosis in the cerebral cortex; neuronal depletion; gliosis.	NA	Brain damage due to insulin overdose.
Junge et al. 2000 [23]	1	68 M	No	Insulin + Metoprolol	SC	Worked as physician; suffered from depression; three syringes containing trace of human insulin were found at the crime scene.	NA	Suicide	Three vital subcutaneous IMs (navel); acute right pulmonary artery embolism (not attached to the wall); lower limb thrombi.	↑ insulin levels in serum (1848.8 μU/mL), CSF (6.9 μU/mL), and VH (0.4 μU/mL); ↓ blood C peptide levels (0.5 μU/mL).	Insulin overdose (with PAE from a lower limb thrombus).
Iwase et al. 2001 [24]	1	25 M	NA	Insulin	IV	Killed by his lover; she also injected 100 mL of air intravenously.	NA	Homicide	IM at the arm; no remarkable findings.	Blood insulin levels 54 μU/mL; blood C peptide levels 166 pmol/L; blood glucose levels 25.4 mmol/L.	Insulin overdose and air embolism.
Uezono et al. 2002 [25]	1 **	5 F	No	Insulin Aspart + Glibenclamide + Benzodiazepines	SC + oral	Arrived dead at the ED; her father was arrested for her death.	↓ blood glucose (1 mmol/L) and C peptide (ng/mL) levels; ↑ blood insulin (694 μU/mL) and glibenclamide (127 ng/mL) levels.	Homicide	Mild pulmonary and renal congestion.	Blood glucose levels were 14.21 mmol/L; ↓ blood C peptide levels (0.5 μU/mL); ↑ blood insulin (295 μU/mL) and glibenclamide (127 ng/mL) levels; ↓ blood C peptide levels (0.5 μU/mL).	Insulin overdose.
Walsh et Sage 2002 [26]	1	57 M	Yes DM2	Insulin	Unspecified	Suffered from depression; CAD; AH; PVD with foot ulcer.	NA	Suicide	Severe CAD.	↑ VH insulin levels (9173.4 μU/mL); hydrocodone + acetaminophen + propoxyphene in blood.	Insulin overdose.
Rao et al. 2006 [27]	1 **	55 M	Yes DM2	Insulin (human mixtard + isophane) + glipizide (tablets)	SC + oral	Worked as physician; suffered from depression; admitted to the ED with diagnosis of hypoglycaemic encephalopathy; he died the day after.	At ED admission, blood glucose levels were 2.72 mmol/L.	Suicide	Vital IMs in the antecubital region of the upper limb; brain congestion and edema; pulmonary congestion; liver steatosis; acute tubular necrosis with diabetic nephropathy.	Negative.	Insulin and glipizide overdose.
Tanenberg et al. 2010 [28]	1	23 M	Yes DM1	Insulin	SC (pump)	An episode of hypoglycemic seizure three weeks before, then found dead in his bed; pump memory indicated that he self-administered five boluses of insulin in 2.5 h.	Finger-stick blood glucose levels were 3.39 mmol/L.	Accidental	No remarkable findings.	↓ VH glucose levels (1.39 mmol/L).	Insulin overdose.
Thevis et al. 2012 [29]	1	55 F	No	Insulin	Unspecified	Hospitalized due to cardiac and hepatic issues; after five weeks, sudden drop of blood glucose levels; deceased after four days.	Blood glucose levels 0.11 mmol/L; blood insulin levels 5551 mU/L; C peptide levels normal.	Homicide	No remarkable findings.	VH insulin levels were 1.0 ng/ml	Insulin overdose.
Hess et al. 2013 [30]	1 *	62 F	No	Insulin Lispro	Unspecified	Her husband suffered from DM; she used his insulin.	NA	Suicide	IM.	↑ insulin levels in the soft tissues under the IMs (>500 uU/g); in the VH (103 μU/mL); in muscles (373 uU/g); and in kidneys (384 uU/g).	Insulin overdose.
Behera et al. 2015 [31]	1	30 F	No	Insulin Lispro	IV	Worked as physician; suffered from depression; an intravenous cannula was found on the left wrist; two empty vials (3 mL each) of insulin 100 IU/mL were found at the crime scene.	At ED admission, blood glucose levels were 1.11 mmol/L.	Suicide	Vital IM over the left wrist joint; Visceral congestion and edema; cerebral petechias, intra-alveolar hemorrhages; micro and macro vesicular steatosis.	↓ VH (0.56 mmol/L) and CSF (2.22 mmol/L) glucose levels.	Insulin overdose.
Sunderland et al. 2016 [32]	1	29 F	Yes DM1	Insulin Glargine (3000 U) + Aspart (260 U)	Unspecified	Suffered from depression; empty insulin pens were found at the crime scene.	NA	Suicide	No remarkable findings.	↓ serum (2.0 mmol/L) and VH (<1.1 mmol/L) glucose levels; ↓ blood C peptide levels (<2.2 μU/mL); ↑ blood insulin levels (total insulin 5.2 μU/mL; aspart insulin 4000 μU/mL).	Insulin overdose.
Tong et al. 2016 [33]	4	27 M	NA	Insulin (Gansulin 800 IU)	SC + IV	The murderer (nurse) injected 200 IU insulin into the victim’s arm. After the man lost consciousness, she injectcted the remainder intravenously at the wrist.	NA	Homicide	IM at the left upper arm and left opisthenar; astrocyte proliferation; necrotic neurons; cerebral edema; necrotic renal tubule epithelium; edema hepatocytes; pulmunary hemorrage; lung edema/congestion.	↓ blood glucose levels (12.16 mmol/L); blood insulin levels were 2.14 mU/L.	Insulin overdose.
		27 M	NA	Insulin (Humulin 400 IU)	SC	The murder (nurse) deceived the victim into drinking a cup of water laced with clonazepam. Once he was asleep, she injected him with insulin.	NA	Homicide	IMs at the right abdomen and elbow; astrocyte proliferation; necrotic neurons; cerebral edema; necrotic renal tubule epithelium; renal granular cast; edema hepatocytes; pulmunary hemorrage; lung edema/congestion; positive immunofluorescence staining of insulin in the injection sites.	↓ blood glucose levels (9.08 mmol/L); ↓ VH glucose levels (0.85 mmol/L); blood insulin levels were 0.2 mU/L.	Insulin overdose.
		55 M	NA	Insulin (Novolin > 1500 IU)	SC	The murder and the victim had divorced, but they still cohabited. The woman was an IDDM patient, and after deceiving the victim into taking a drink laced with clonazepam, she injected him with insulin.	NA	Homicide	IM at the arm and middle abdomen; astrocyte proliferation; necrotic neurons; cerebral edema; necrotic renal tubule epithelium; renal granular cast; edema hepatocytes; focal necrosis of liver; necrotic liver tissue; pulmunary hemorrage; lung edema/congestion; positive immunofluorescence staining of insulin in the injection sites.	Blood insulin levels 0.2 mU/L.	Insulin overdose.
		86 F	NA	Insulin (Aspart 1200 IU)	SC	The murderer injected a lethal amount of insulin into her mother (IDDM patient). She then attempted suicide but failed.	NA	Homicide	Abdominal IM; astrocyte proliferation; necrotic neurons; cerebral edema; necrotic renal tubule epithelium; renal granular cast; edema hepatocytes; focal necrosis of liver; necrotic liver tissue; pulmunary hemorrage; lung edema/congestion; positive immunofluorescence staining of insulin in the injection sites.	↓ blood glucose levels 11.4 mmol/L; ↓ VH glucose levels 0.9 mmol/L; blood insulin levels were 2.5 mU/L.	Insulin overdose.
Ikeda et al. 2018 [34]	1	55 F	No	Glibenclamide + amlodipine	Oral	Her husband suffered from DM; another episode of hypoglycemia some days before the suicide.	NA	Suicide	Epicardial petechiae; myocardial edema; lung edema and intralveolar hemorrhages; visceral edema; insulin immunostaining of the pancreatic islets was normal.	↓ glucose levels in blood and fluids (0.06–0.28 mmol/L); in VH and CSF (0.06 mmol/L); with no increase in insulin (1.34–10.4 μg/mL) or C peptide (0.25–1.41 ng/mL) levels; lactate levels in VH and CSF were 1 mmol/L; ↑ blood glibenclamide (54 and 81 ng/mL) and amlodipine (190 and 72 ng/mL) levels; Glibenclamide and amlodipine were also detected in PCF, BMA, urine, and stomach content.	Glibenclamide and amlodipine overdose.
Bugelli et al. 2019 [35]	1	68 M	No	Insulin regular	Sublingual (supposedly)	Worked as paramedical personnel; suffered from depression; 2 empty vials (10 mL each) of Humulin R insulin were found at the crime scene.	NA	Suicide	Pulmonary edema; focal alveolar hemorrhages; visceral congestion.	↓ glucose (<0.06 mmol/L); ↑ potassium (>10 mEq/L); ↑ insulin (61.11 mU/L and 74.23 mU/L); and ↓ C peptide levels in VH.	Insulin overdose.
Shah et al. 2020 [36]	1	51 M	Yes DM1	Insulin	SC (pump)	He disabled the auto mode of his insulin delivery device and dosed about 180 U of insulin. Died after two days of hospitalization.	NA	Suicide	NA	NA	Insulin overdose.
Bottinelli et al. 2021 [37]	13	56.4 (range 32–82), 8 M and 5 F	10 yes, 3 no	Insulin (various types)	SC	-	NA	6 suicide; 7 accidental	NA	In 11 out of 13, Insulin detected in at least one organ (in 4 cases, PMI > 48 h).	Insulin overdose.
Nishihama et al. 2021 [38]	1	65 F	Yes DM1	Insulin	SC (pump)	She mistakenly disabled the pump.	NA	Accidental	NA	NA	Insulin overdose.
Stephenson et al. 2022 [39]	40	49.5, 19 F and 21 M	Suicide: 16 yes, 3 no, 10 unspecified; Accidental: 6 yes, 2 no, 3 unspecified	Insulin	1 IV, 22 SC, 17 unspecified	-	-	29 suicide; 11 accidental	External examination showed IMs in 23 cases (19 SC at the abdomen, 5 SC at the thigh, 1 SC at the hand, 1 SC at the antecubital fossa and 1 IV in arm); histopathological exam showed lung edema and congestion in 28 cases, early focal bronchopneumonia in 8 cases, early hemorrhagic acute pneumonia in 1 case, and neuronal necrosis in 17 cases.	Blood insulin detection in 29 cases, blood C peptide detection in 26 cases.	Insulin overdose in 28 cases; drowning and insulin overdose in 1 case; insulin and other drugs overdose in 5 cases; brain injuries due to insulin toxicity in 5 cases; acute broncopneumonia due to insulin toxicity in 1 case.

ACA indicates anterior communicating artery; AH, arterial hypertension; BMA, bone marrow aspirate; CAD, coronary artery disease; CSF, cerebrospinal fluid; DM, diabetes mellitus; ED, emergency department; HbA1c, glycated hemoglobin; IM, injection mark; IV, intravenous; NA, not available; NPH, neutral protamine hagedorn; PAE, pulmonary artery embolism; PCF, pericardial fluid; PMI, post-mortem interval; PVD, peripheral vascular disease; SC, subcutaneous; VH, vitreous humor. * These articles also show cases of survivors or cases of death due to other drugs. These cases were excluded from our review. ** These articles show cases with joint administration of insulin and oral hypoglycemic agents.

**Table 2 biomedicines-10-02823-t002:** This table shows a summary of the main characteristics of the cases included in the review.

Characteristics		Suicide	Homicide	Accidental	Total
Sex	Male	40	7	16	63
	Female	31	6	9	46
Age (yrs) *	0–18	0	1	0	1
	19–30	6	5	1	12
	31–50	9	2	5	16
	>50	14	5	8	27
	Unspecified	53	0	0	53
Type of drug	Insulin	69	11	24	104
	OHA	0	3	0	3
	Insulin + OHA	1	1	0	2
Route of administration **	Intravenous	6	3	0	9
	Subcutaneous	55	8	21	84
	Oral	2	3	0	5
	Sublingual	1	0	0	1
	Unspecified	17	2	4	23
Previous diagnosis of diabetes mellitus	Yes	37	1	20	58
	No	18	6	2	26
	Unspecified	16	6	3	25
Health care worker	Yes	14	1	0	15
	No	57	12	25	94

OHA indicates oral hypoglycemic agent. * Cases with unspecified age or when only the mean age was provided were not included in this table. ** A more detailed description of the routes of administrations is provided in Table 3.

**Table 3 biomedicines-10-02823-t003:** This table is an extension of a section of Table 3. It shows the details about injection sites presented in individual cases when the intravenous or the subcutaneous route was performed.

Route of Administration	Site of Injection	Suicide	Homicide	Accidental	Total
Intravenous	Arm/forearm	2	1	0	3
	Antecubital region	1	0	0	1
	Wrist	1	1	0	2
	Unspecified	2	1	0	3
Subcutaneous	Chest	1	0	0	1
	Abdomen	15	3	6	24
	Thigh	8	0	1	9
	Arm/forearm	2	3	0	5
	Antecubital region	3	0	0	3
	Elbow	0	1	0	1
	Hand	2	0	0	2
	Buttock	1	2	0	3
	Leg	1	0	0	1
	Pump	1	0	2	3
	Unspecified	21	1	12	34

**Table 4 biomedicines-10-02823-t004:** This table shows the toxicological examinations performed in the included cases.

Toxicological Analyses	Insulin (*n* = 76)	OHA (*n* = 1)	Insulin + OHA (*n* = 2)
Blood insulin	53	1	2
Blood C peptide	30	1	1
Vitreous humor glucose	19	1	0
Injection site insulin	17	-	0
Liquor glucose	14	1	0
Organ insulin	14	-	0
Liquor insulin	13	-	0
Blood HbA1c	12	0	0
Blood glucose	10	1	1
Other drugs (one or more matrices)	7	1	1
Vitreous humor insulin	5	0	0
Urine glucose	2	0	0
Syringe content (insulin)	2	0	0
Vitreous humor potassium	1	0	0
Vitreous humor C peptide	1	0	0
Blood glipizide	-	0	1
Blood glibenclamide	-	1	1

HbA1c indicates glycated hemoglobin; OHA, oral hypoglycemic agents.

**Table 5 biomedicines-10-02823-t005:** This table shows the post-mortem findings in the included cases.

Autoptic Data		Insulin (*n* = 79)	OHA (*n* = 3)	Insulin + OHA (*n* = 2)
External examination	Injection mark	53	0	1
	Evidence of vomitus	2	0	0
	Fingernail cyanosis	1	0	0
	Petechiae in hypostatic areas	1	0	0
Internal examination	Brain edema	17	0	1
	Brain congestion	0	0	1
	Brain petechial hemorrhages	0	0	1
	Lung froth	1	0	1
	Pulmonary edema	44	0	1
	Pulmonary congestion	33	0	2
	Pleural petechial hemorrhages	1	0	0
	Epicardial petechiae	1	0	0
	Heart petechiae	1	0	0
	Heart edema	1	0	0
	Visceral edema	1	0	0
	Visceral congestion	12	0	1
	Unspecified/unremarkable findings	32	0	0
Histology	Glial cell proliferation (gliosis)	6	0	0
	Neuronal depletion/necrosis	23	0	0
	Cerebral perivascular hemorrhages	1	0	0
	Pulmonary edema with intra-alveolar hemorrhages	4	0	0
	Early focal bronchopneumonia	8	0	0
	Early hemorrhagic acute pneumonia	1	0	0
	Focal liver necrosis	2	0	0
	Hepatocytes edema	4	0	0
	Focal renal necrosis	4	0	0
	Injection site: fresh hemorrhage	1	0	0
	Injection site: interstitial edema	1	0	0
	Injection site: immunofluorescence staining of insulin	3	0	0

OHA indicates oral hypoglycemic agents. Autopsy data related to conditions other than hypoglycemia were excluded (for example, signs of drowning), as were data from autopsy performed in long-term hospitalized patients.

## Data Availability

Not applicable.

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
