# Peer review of "Insulin and Oral Hypoglycemic Drug Overdose in Post-Mortem Investigations: A Literature Review"

_biomedicines, 2022, doi:10.3390/biomedicines10112823_

Round 1

Reviewer 1 Report

In this  review the authors tried  relate the Insulin and oral hypoglycemic drug overdose in post-mortem investigations. The authors had tried to review the  published literature from 1958 till 2022. The references and literature presented are up to date however, there is no novelty in this review. There many published review articles in same topic and hence this review will not add much scientific information to the current body of scientific knoweldge. Here are some studies with similar literature review.

1- Is insulin intoxication still the perfect crime? Analysis and interpretation of postmortem insulin: review and perspectives in forensic toxicology, Critical Reviews in Toxicology, 50:4, 324-347, DOI: 10.1080/10408444.2020.1762540

2-The Determination of Insulin Overdose in Postmortem Investigations June 2016Academic Forensic Pathology 6(2):174-183. DOI:10.23907/2016.019

3-The Other Face of Insulin—Overdose and Its Effects. Toxics 2022, 10, 123. https://doi.org/ 10.3390/toxics10030123

and many more.  The authors have to consider these publication as well while reviewing the topic.

Moreover, the conclusion must have to be revised and conclude main findings of the study rather then reporting the results again

The review can be re submitted after revision. 

Author Response

Dear Reviewer, thank you for your comment.

Our work focused on the research of case reports in the literature from 1958 to 2022 and it is about deaths due to hypoglycemic drugs overdose. The papers you cited certainly bring improvements, for this reason we took a cue and added them to the discussion. We have revised the conclusion following your advice. We thank you for your observations.

Reviewer 2 Report

I congratulate the authors for their research. The research was very well conducted and brings relevant data on a little explored subject. Below, I cite some considerations to improve the manuscript.

1 - The abstract is well written. In my opinion the authors should cite which countries or continents the study approached.

2 - The authors could better detail from which country are the data reported in the lines 40 to 53? 

3 - Line 73 - in my opinion, I believe it would be interesting to cite the countries or continents where the studies and case reports were carried out.

4 - Figure 1 illustrates the research design very well.

5 - Lines 200 to 208: In my opinion, it was not clear how the drugs mentioned in the studies were responsible for the crimes committed by their users.

Author Response

Dear Reviewer, thank you for your appreciations.

Although some articles mentioned the Countries where the case took place, on the other hand in many it is not explicated. However, we added a few lines specifying that the most of the papers are from Western countries (lines 55-56 and 90-91)

As you suggested, we implemented lines 200-208 (now 212-218).

Reviewer 3 Report

Authors briefly reviewed the major postmortem investigations in cases of a suspected hypoglycemic drug overdose. This is an interesting review and can help forensic pathologists to suspect hypoglycemic drug overdose when the cause of death is not clear, but there are some details that should be reviewed by the authors.

-In Methods section. Exclusion criteria is not well defined.

- I suggest to expand the conclusion and propose guidelines for future research.

-Be specific when referring to studies and use authors names and be mindful of the point that you are trying to make. It appears that there is very little critical appraisal throughout the document.

-The authors could add a scheme figure showing the potential mechanisms of the major postmortem investigations in cases of a suspected hypoglycemic drug overdose.

Author Response

Dear Reviewer, thank you for your advice.

We added a few lines regarding the exclusion criteria, in addition to the inclusion criteria we have already mentioned in the materials and methods section (lines 74-77).

In the discussion section, we added other references and proposed an intuitive flowchart that the forensic pathologists could follow in cases of the suspect of death due to insulin or OHA overdose.

We have revised the conclusion following your advice, proposing guidelines for future research.

Round 2

Reviewer 1 Report

The authors have discussed the findings from the suggested publications in the revised manuscript. However, the conclusion is still very big, it could have been shortened.

The MS can be accepted without further revision

Reviewer 3 Report

Please move "In Table 1, a brief description of the studies included in this review is reported."this sentence to line 92. and delated the sentence ( line 97). The authors have addressed some of my comments, I think the authors have done a good job.